# Synthetic Peptides Elicit Humoral Response against Porcine Reproductive and Respiratory Syndrome Virus in Swine

**DOI:** 10.3390/vaccines12060652

**Published:** 2024-06-11

**Authors:** Francisco Perez-Duran, Fernando Calderon-Rico, Luis Enrique Franco-Correa, Alicia Gabriela Zamora-Aviles, Roberto Ortega-Flores, Daniel Durand-Herrera, Alejandro Bravo-Patiño, Ricarda Cortes-Vieyra, Ilane Hernandez-Morales, Rosa Elvira Nuñez-Anita

**Affiliations:** 1Facultad de Medicina Veterinaria y Zootecnia, Universidad Michoacana de San Nicolas de Hidalgo, Km. 9.5 S/N Carretera Morelia-Zinapecuaro, La Palma, Tarimbaro CP 58893, Mexico; 0618713g@umich.mx (F.P.-D.); 1028153g@umich.mx (F.C.-R.); 1837980k@umich.mx (L.E.F.-C.); 1106849c@umich.mx (A.G.Z.-A.); 0840020c@umich.mx (R.O.-F.); daniel.durand@umich.mx (D.D.-H.); alejandro.bravo@umich.mx (A.B.-P.); ricarda.cortes@umich.mx (R.C.-V.); 2Laboratorio de Investigacion Interdisciplinaria, Escuela Nacional de Estudios Superiores Unidad Leon, Universidad Nacional Autonoma de Mexico, Blv. UNAM No. 2011, Leon CP 37684, Mexico; ihernandez@enes.unam.mx

**Keywords:** PRRS, B-cells, CD2+, CD21+, GP5

## Abstract

The aim of this study was to analyze the immunogenic response elicited in swine by two synthetic peptides derived from GP5 to understand the role of lineal B epitopes in the humoral and B-cell-mediated response against the porcine reproductive and respiratory syndrome virus (PRRSV). For inoculation, twenty-one-day-old pigs were allocated into six groups: control, vehicle, vaccinated (Ingelvac-PRRSV, MLV^®^), non-vaccinated and naturally infected, GP5-B and GP5-B3. At 2 days post-immunization (dpi), the GP5-B3 peptide increased the serum concentrations of cytokines associated with activate adaptive cellular immunity, IL-1β (1.15 ± 1.15 to 10.17 ± 0.94 pg/mL) and IL-12 (323.8 ± 23.3 to 778.5 ± 58.11 pg/mL), compared to the control group. The concentration of IgGs anti-GP5-B increased in both cases at 21 and 42 dpi compared to that at 0 days (128.3 ± 8.34 ng/mL to 231.9 ± 17.82 and 331 ± 14.86 ng/mL), while IgGs anti-GP5-B3 increased at 21 dpi (105.1 ± 19.06 to 178 ± 15.09 ng/mL) and remained at the same level until 42 dpi. Also, antibody-forming/Plasma B cells (CD2+/CD21−) increased in both cases (9.85 ± 0.7% to 13.67 ± 0.44 for GP5-B and 15.72 ± 1.27% for GP5-B3). Furthermore, primed B cells (CD2−/CD21+) from immunized pigs showed an increase in both cases (9.62 ± 1.5% to 24.51 ± 1.3 for GP5-B and 34 ± 2.39% for GP5-B3) at 42 dpi. Conversely the naïve B cells from immunized pigs decreased compared with the control group (8.84 ± 0.63% to 6.25 ± 0.66 for GP5-B and 5.78 ± 0.48% for GP5-B3). Importantly, both GP5-B and GP5-B3 peptides exhibited immunoreactivity against serum antibodies from the vaccinated group, as well as the non-vaccinated and naturally infected group. In conclusion, GP5-B and GP5-B3 peptides elicited immunogenicity mediated by antigen-specific IgGs and B cell activation.

## 1. Introduction

Porcine reproductive and respiratory syndrome (PRRS) is an endemic disease in all countries with large pig holdings, and it is likely the most economically significant disease for the global swine industry [1,2,3,4]. This disease affects pigs of all ages and is characterized by reproductive failures in sows during late-term gestation, including high rates of abortions, mummified fetuses, and post-weaning mortality due to the birth of weak piglets, with perinatal mortality reaching up to 70%. Additionally, it causes respiratory disease in both piglets and adult swine [3,5,6]. PRRSV has shown a high capacity for infection and transmission through the oronasal and reproductive routes [7].

The disease is caused by the similarly named Porcine Reproductive and Respiratory Syndrome Virus (PRRSV), within the family *arteriviridae*, in the genus *Betaarterivirus*, order *Nidovirales* [3,8,9]. Two genotypes of PRRS virus have been reported: PRRSV type 1, or European; and PRRSV type 2, or American [10]. However, there is a significant genetic variability among different PRRSV isolates within each genotype [11]. PRRSV is an enveloped RNA virus, measuring 45–70 nm in diameter [12]. Its viral genome consists of a polyadenylated, single-stranded, non-segmented, positive-sense, polycystronic RNA molecule ranging from 15.1 to 15.5 kb in size and contains 11 open reading frames (ORFs), including ORF1a, ORF1b, ORF2a, ORF2b, ORF3-7, ORF5a and ORF1aTF. Among them, ORF1a, ORF1b and ORF1aTF account for 80% of the viral genome. The translated polypeptides pp1a and pp1ab (encoded in ORF1a and ORF 1b) are cleaved into 15 Non-Structural Proteins (NSPs), including NSP1α, NSP1β, NSP2-related proteins (NSP2N, NSP2TF, and NSP2), and NSP3–NSP12, which are involved in viral replication [13]. ORF2a, ORF2b, ORF3, ORF4, ORF5, and ORF5a encode six structural envelope-related proteins, namely, GP2a, GP2b, GP3, GP4, GP5, and GP5a, respectively [14,15,16,17]. Structural proteins are glycoproteins (GPs) embedded on the lipid envelope, forming protein complexes that provide stability to the viral particle and participate in the recognition and internalization of the virus to its target cell [15,18]. The nonglycosylated matrix (M) protein encoded by ORF6 and the nucleocapsid (N) protein encoded by ORF7 are dominant structural proteins with strong immunogenicity [19].

Vaccination is the main prevention method, and several vaccine formulations have been developed against PRRSV; some of these have shown low efficacy, while others are still in the experimental phase [15]. Current commercial vaccines against PRRSV include various types of single and combined modified live viruses (MLV) from PRRSV 1 to PRRSV 2, as well as inactivated viruses, primarily PRRSV 1 but also PRRSV 2 [20,21]. The commercially available MLVs vaccines fail to provide sufficient heterologous protection, as they typically induce weak innate and humoral responses, and inadequate T cell responses [22,23,24,25,26]. For instance, the antibody protection following vaccination is low and may result in antibody-dependent enhancement, facilitating virus entry [27]. Inactivated virus vaccines have shown poor efficacy, such as poor immune effects on heterologous strains, a lack of detectable PRRSV-specific antibody production, and an absence of a cell-mediated immune response [25,27,28,29,30,31,32]. The high genetic and antigenic diversity presents a significant impediment to developing an effective vaccine to control PRRS. In addition, it is known that, in some cases, attenuated vaccine viruses [21] are associated with the appearance of severe outbreaks due to the vaccine virus reverting to a pathogen status [33]. Therefore, research into safe and effective antigens is necessary to induce an effective immune response.

Structural glycoprotein GP5 has been reported as a promising candidate for vaccine development as it is the main target of neutralizing antibodies due to the presence of epitopes recognized by B cells. Moreover, GP5 is highly immunogenic, making it significant in the diagnosis, prevention, and control of PRRSV [16,34,35,36,37,38,39]. Effective anti-PRRSV immunity may be attained by exposing immunogenic epitopes to induce efficient innate and adaptive immune responses mediated by specific antibodies, cytokines, and T-cell responses [17,40]. Compared to commercial vaccines, peptide vaccines do not contain nucleic acid substances; therefore, they are considered safer [41]. Peptides contain only B or T epitopes that induce a specific response [27,41]. Likewise, adjuvants can enhance T or B cells’ responses by enhancing phagocyting activity and secreting a variety of cytokines that boost the specific immune response to the vaccine [32,42,43]. The goal of this work was to determine the immunogenic response elicited in swine by two synthetic peptides derived from the GP5 of PRRSV to understand the role of B epitopes in the immune response against PRRSV and to support the further development of these peptides as vaccine candidates.

## 2. Materials and Methods

### 2.1. Peptides

The GP5-B and GP5-B3 peptides contain epitopes that match the peptide sequence within the ectodomain of the GP5 protein (residues 30 to 62) from PRRSV type 2 (Sequence ID: UTS56108.1). Our research group has previously studied the GP5-B peptide, while the GP5-B3 peptide encompasses two epitopes previously reported by Vashit et al., 2008 [44]. Both peptides demonstrated high antigenicity in silico, according to the ImmunoEpitope DataBase IEDB and CCL Main Workbench 20.0 (unpublished data).

The peptides were chemically synthesized through solid-phase synthesis by GenScript (Piscataway, NJ, USA). The GP5-B peptide sequence, ASNDSSSHLQLIYNLTLCELNGTDWLANKF (30 aa), and the GP5-B3 peptide sequence, SSSNLQLIYNLTTPVTRVSAEQWGRPC (27 aa), exhibited 95% and 98% purity, respectively. The GP5-B3 peptide was modified by adding a cysteine at the terminal carboxyl end to facilitate binding with a carrier protein. Both peptides were resuspended in dimethyl sulfoxide (DMSO) and adjusted to a stock solution concentration of 1 mg/mL.

The sequences of the GP5-B and GP5-B3 peptides were compared with those reported for PRRSV types 1 and 2, as well as high pathogenic strains, using the Protein BLAST program. The results revealed query coverage for GP5-B of 93–100% for PRRSV type 1, 96–100% for PRRSV type 2, and 94–100% for HP-PRRSV. For GP5-B3, the query coverage was 77–96% for PRRSV type 1, 92–96% for PRRSV type 2, and 90–96% for HP-PRRSV.

### 2.2. Experimental Design, Animals, and Housing

This research was conducted on piglets procured from a certified farm (El Dapo farm, which conducts Good Livestock Practices under the Mexican official norm which NOM-033-ZOO-1995), where periodic testing ensured a negative status for PCV2 and PRRSV. Piglets, 21 days old, weaned, of indistinct sex, and interbreed large white x pietrain, were selected randomly from six different litters. All piglets received iron supplementation and vaccines as described previously [45].

Twenty-eight piglets were moved to alternative experimental stockyards larger than 33.6 m^2^ in “La carreta” farm. Each pig was ear-tagged and distributed randomly across two stockyards. Using the GraphPad-Random number tool, each animal was randomly allocated to one of four treatment groups, with each group comprising seven individuals.

Controls in the study: animals, samples, materials, and procedures are described in Table 1 and Figure 1. Pigs were visually monitored daily and weighed every 21 days and maintained as previously described, following the applicable guidelines under the approved study protocol CICUMSNH-A101-FMVZ [45].

Additionally, an immunoreactivity assay was performed using sera from two groups: naturally infected pigs and vaccinated pigs (Table 1). The sera from seven naturally infected pigs (N = 7), without prior PRRSV vaccination, were collected from “El Limon” farm. The sera from seven (N = 7) vaccinated pigs were collected from “El Dapo” farm. The piglets were immunized intramuscularly by restraining their legs (day 0) with a 2 mL/piglet single dose of the Ingelvac-PRRSV, MLV^®^ (containing virus ATCC VR 2332 to a concentration of 104.9 DICC_50_).

### 2.3. Blood and Serum Sampling and Isolation of Peripheral Blood Mononuclear Cells

At 2 and 42 days post immunization (dpi), 15 mL blood samples were drawn via the jugular vein, with and without EDTA (Vacutainer, BD, NJ, USA) from the control, vehicle, GP5-B, and GP5-B3 peptide groups at ‘El Dapo farm’. The samples were used no later than 3 h after collection. The protocol to isolate Peripheral Blood Mononuclear Cells (PBMC) was carried out as described previously [45].

Serum samples from eighty-day-old vaccinated pigs were collected at 42 dpi, while the serum samples from naturally infected pigs were obtained at 16 weeks of age in July 2023.

### 2.4. Quantification of Cytokine Concentrations in Serum and Antibody Detection

The cytokine concentration in the serum was determined as described before [45]; the pre-configurated cytokine panel included IFN alpha, IFN gamma, IL-1 bet], IL-10, IL-12/IL-23p40, IL-4, IL-6, IL-8 (CXCL8), and TNF alpha.

Peptide-specific antibody analysis was performed using an indirect Enzyme-Linked Immunosorbent Assay (ELISA) to quantify immunoglobulins G (IgGs) against each of the GP5-B and GP5-B3 peptides. The commercial Peptide Coating Kit (Takara, Shiga, Japan) was used for the ELISA test. Following the manufacturer’s protocol, the 96-well plate was coated with each peptide at a concentration of 4 µg/mL for 2 h at room temperature (RT), and then the plate was blocked with a block solution for 2 h (solution provided in the kit) at RT. The plates were washed three times with distilled water. Then, serum samples from piglets were diluted 1:100 in phosphate-buffered saline-T (PBS containing Tween-20 at 0.05%) containing 0.3% gelatin, and added to the plate. After incubating overnight at 4 °C, three washes were performed and the plates were incubated with the goat anti-porcine-IgG (H+L)-HRP antibody (SouthernBiotech, Birmingham, AL, USA) in dilution 1:1000 for 2 h at RT. The plate was washed three times with PBS-T, after which the substrate was added, and the plate was read at 15 min. The reading was carried out at OD_450_ in a plate microreader (Bio-Rad iMARK microplate reader). The results are shown in [ng/mL] according to the IgG’s calibration curve.

### 2.5. Immunoreactivity of Peptides against Serum Antibodies

The immunoreactivity assay was performed using an indirect ELISA with the Peptide Coating Kit, as described above. Immunoreactivity was assessed in serum samples from both the vaccinated group and the naturally infected (non-vaccinated) group. The results are presented in [ng/mL], based on the IgG’s calibration curve.

### 2.6. Flow Cytometry

Cytometry was performed with approximately 1 × 10^6^ cells/mL of PBS per sample, which was evaluated in the flow cytometer (Attune NxT acoustic focusing cytometer, Thermo Fisher Scientific, Carlsbad, CA, USA). The cytometer was equipped with a violet laser (405 nm) and a blue laser (488 nm). The cells were stained with antibodies coupled to fluorochromes to detect specific surface markers of antibody-secreting B cells. The analysis targeted three subpopulations of B cells: IgM+/CD2+/CD21+ (naïve); CD2+/CD21− (plasmocytes) and CD2−/CD21+ (primed). Surface markers were used for triple staining, enabling primary antibodies to form combinations of goat-anti-pig IgM-FitC (MBS224956, MyBioSource, San Diego, CA, USA) with anti-pig-CD2-PECy7 (RPA-2.10, eBioscience, San Diego, CA, USA) and anti-pig-CD21-PE (BB6-11C9.6, Invitrogen, Carlsbad, CA, USA).

For the analysis, 10,000 individual event packages were captured within the singlets gate, and a dot plot contrasting forward scatter-A with forward scatter-H was generated to exclude doubles. Additionally, a linear dot plot comparing side scatter to linear forward scatter was utilized to delineate the lymphocytes among the singlets. Another panel from the lymphocytes compared BL1-IgM+ cells against the linear scatter side. Finally, a quad-rant panel analysis of IgM+ identified four subpopulations of IgM+ B cells containing CD2+/CD21− (Q1), CD2+/CD21+ (Q2), CD2−/CD21+ (Q3) and double-negative CD2−/CD21− cells (Q4), CD2−/CD21−, CD2+/CD21+CD2+/CD21− and CD2−/CD21+. The quadrant panel was delimited from Fluorescent Minus One (FMO) controls. The results from the subpopulations are presented as the percentage of cells that tested positive.

### 2.7. Statistical Analysis

The data represent three different measurements for each animal and are presented as means ± standard error of the mean (SEM). Differences between groups were estimated using a one-way ANOVA test with Tukey’s pos hoc analysis. A *p*-value of ≤0.05 was considered statistically significant. The GraphPad Prism software (version 9.1.1, GraphPad Software, San Diego, CA, USA) was used. Flow cytometry data were analyzed using the FlowJo V. X software (FlowJo_v10.7.2) (BD^®^).

## 3. Results

### 3.1. Body Weight Monitoring after Immunization

Body weight was monitored from day 0 to day 42. All animals maintained a constant body condition and gained weight throughout the study (Figure 1). There are non-significant differences between the control and peptide-immunized groups. Furthermore, no adverse reactions to immunization were observed in any of the four groups.

### 3.2. GP5-B3 Peptide Increased the Serum Concentration of Proinflammatory Cytokines

Cytokine concentrations were quantified in serum samples collected at 2 dpi to determine the proinflammatory state. Pigs immunized with GP5-B3 exhibited significantly higher serum concentrations of IL-1β and IL-12; IL-1β from 1.15 ± 1.15 to 10.17 ± 0.94 pg/mL with respect to the control group and from 2.31 ± 1.49 to 10.17 ± 0.94 pg/mL with respect to the vehicle group. IL-12 increased from 323.8 ± 23.3 to 778.5 ± 58.11 pg/mL with respect to the control group and from 483.2 ± 50.56 to 778.5 ± 58.11 pg/mL with respect to the vehicle group (Figure 2A,B).

There were no significant differences in cytokine serum concentrations between the GP5-B group and the control group. This observation implies that the GP5-B3 peptide might trigger a proinflammatory response following immunization. To assess the immunomodulatory effects at the onset of antigen processing and presentation, cytokine levels were measured two days post-immunization.

### 3.3. GP5-B and GP5-B3 Induce Specific IgG-Mediated Response upon Immunization

The group immunized with the GP5-B peptide showed a significant increase in anti-GP5-B IgGs at 21 and 42 dpi compared to day 0 (pre-immune) from 128.3 ± 8.34 to 231.9 ± 17.82 and 331 ± 14.86 ng/mL, respectively. Additionally, a significant increase was observed at 42 dpi (331 ± 17.86 ng/mL) compared to 21 dpi (231.9 ± 17.82 ng/mL). In the case of the GP5-B group, a sustained increase in IgGs concentration was observed (Figure 3A). Conversely, in the group immunized with the GP5-B3 peptide, a significant induction of anti-GP5-B3 IgGs was observed only at 21 dpi relative to day 0 from 105.1 ± 19.06 to 178 ± 15.09 ng/mL. No induction of IgGs was observed at 42 dpi; thus, the IgGs’ increase observed at 21 dpi remained at the same concentration until 42 dpi (Figure 3B). The data demonstrate a clear IgG-mediated response following immunization with the peptides GP5-B or GP5-B3.

### 3.4. Peptides Prime B Cells and Induce the Generation of Antibody-Secreting Cells

Immunotyping of B cells was performed at 42 dpi, and percentages of IgM cells and subpopulations (CD2+/CD21−; CD2−/CD21+; CD2+/CD21+) were evaluated for four experimental groups: the control, vehicle, GP5-B and GP5-B3 groups. Figure 4 shows the strategy route to analyze B cell subpopulations. The results showed that, in the case of B cells, the groups immunized with GP5-B or GP5-B3 peptides increased the percentage of antibody-forming/plasma B cells (CD2+/CD21−) compared to the control group from 9.85 ± 0.7% to 13.67 ± 0.44 and 15.72 ± 1.27%, respectively, and that of the vehicle group from 10.25 ± 0.4% to 13.67 ± 0.44 and 15.72 ± 1.27%, respectively; however, no differences were observed between the two peptide-immunized groups (Figure 5A). This suggests that the peptides GP5-B and GP5-B3 induce the generation of antibody-secreting cells.

On the other hand, the primed B cells’ subpopulation CD2−/CD21+ showed a statistically significant increase in the groups immunized with the peptides GP5-B or GP5-B3 compared to the control from 9.62 ± 1.5% to 24.51 ± 1.3 and 34 ± 2.39%, respectively, and vehicle groups from 17.1 ± 1.23% to 24.51 ± 1.3 and 34 ± 2.39%, respectively. Additionally, in the case of the GP5-B3 group, there was a statistically significant increase compared to the group immunized with the GP5-B peptide from 24.51 ± 1.3 to 34 ± 2.39% (Figure 5B), indicating that the GP5-B3 peptide was the most effective inducer of primed B cells.

Finally, the evaluation of naïve B cells (CD2+/CD21+) showed a statistically significant reduction across all experimental groups compared to the control group. However, no differences were observed in the percentage of these cells among the three groups (Figure 5C). This suggests that, in response to peptide immunization, the naïve cell subpopulation was reduced, but this reduction was also seen in response to the BSA carrier in equal proportions from 8.84 ± 0.63% to 5.8 ± 0.4%.

### 3.5. GP5 Epitopes Elicit In Vitro Immunoreactivity in Sera from Naturally Infected and Vaccinated Pigs

Our next question was whether the B epitopes contained in GP5-B or GP5-B3 peptides could be conserved epitopes capable of inducing antibodies during natural infection with PRRSV or after MLV vaccination. Figure 6 illustrates the immunoreactivity of both GP-B and GP5-B3 peptides against serum antibodies from naturally infected and MLV vaccinated animals. The response against the two peptides was significantly higher in the group of vaccinated animals than in those naturally infected. However, the naturally infected group also exhibited significant immunoreactivity against the two peptides compared to the immunized control group. Interestingly, this indicates that the B epitopes in GP5-B or GP5-B3 are conserved in circulating wild viruses.

## 4. Discussion

PRRSV-specific IgG neutralizing antibodies are produced 3–4 weeks after infection [46], which is too late to stop the acute phase of viremia [46]. The use of epitopes that are potential inducers of neutralizing antibodies is a topic of great importance in the research of safe and effective vaccines [46]. However, beyond the monitoring of the antibody response, it has become evident that an understanding of B cell behavior, that is, the cellular biology underlying the antibody response, is crucial for understanding how the immune response functions in the context of vaccination [47]. Consequently, the aim of this study was to analyze the immunogenic response elicited in swine by two synthetic peptides derived from GP5 to understand the role of lineal B epitopes in the humoral and B-cell-mediated response and to support the further development of these peptides as vaccine candidates.

The humoral immune response to PRRSV infection has been frequently studied, especially the one directed to different envelope proteins (GP3, GP4, GP5 or M) or linear epitopes from these proteins [16,24,43,48]. The present study evaluated immunogenic epitopes within the GP5 protein using GP5-B and GP5-B3 synthetic peptides derived from PRRSV-2 that contain linear epitopes recognized by B cells.

Previous studies demonstrate that synthetic peptides containing B cell epitopes induce a specific GP5 specific humoral response and proinflammatory cytokines [34]. This observation is confirmed in the resent study, where immunization of piglets with the GP5-B and GP5-B3 peptides showed a significant increase in the concentration of specific IgGs after 21 days after primary immunization in both cases. However, only the GP5-B group showed a statistically significant increase after secondary immunization, suggesting a strong and sustained response, indicating the presence of active antibody producing cells. Conversely, the GP5-B3 group did not show a specific increase in IgGs after secondary immunization; nevertheless, its IgG levels remained elevated until 42 dpi, indicating the presence of antibody-forming/Plasma B cells [49].

The presence of memory B cells allows antigen-specific antibodies to be rapidly generated when the antigen that induced them comes into contact with these cells a second time, resulting in a faster and stronger humoral response [50]. Pre-immune animals exhibited a basal concentration of specific antibodies against GP5-B peptide. This could be explained by nonspecific binding of antibodies to the ELISA plate or a cross-reaction with the assay´s microarrangement [51,52]. Another potential explanation for these readings at day 0 is the presence of maternal antibodies, which were likely transferred from the mother to the piglets via colostrum. According to Guzman-Bautista et al., 2013 [53], maternal antibodies in pigs can provide protection against respiratory infections through IgG, the predominant isotype in pig breast milk. These maternal antibodies also provide protection through the neutralization of pathogens and the recruitment of innate immunity effector molecules through the Fc domain of antibodies [54]. Thus, it is plausible that maternal antibodies could recognize the epitopes of both peptides given that the mothers of the piglets had been vaccinated.

The activation of B cells correlates with the induction of an antibody-mediated humoral response [55]. The humoral response mediated by B cells was also evaluated through three subpopulations: (i) antibody-forming/plasma B cells (CD2+/CD21−), which are B cells that correspond to B cells circulating within the peripheral blood with the function of secreting antibodies that can potentially recognize the specific antigen (anti-peptide antibody) that stimulated the cell [56,57]. (ii) Primed B cells (CD2−/CD21+), which present their B receptors (BCR) charged with the antigen and remain in circulation waiting to contact with T cells that provide the activation signal [58]. (iii) Naïve B cells (CD2+/CD21+), which circulate through the peripheral blood and have not been exposed to an antigen [56,59]. Thus, this work evaluated the induction of subpopulations of B cells as follows: plasmatic cells (CD2+/CD21−), trained cells (CD2−/CD21+), and naïve cells (CD2+/CD21+). All these subpopulations were analyzed based on the selection of B cells with reduced IgM expression (IgM low).

Natural infection with PRRSV primes B cells and triggers antibody production and memory B cells [60]. Moreover, the expression of PRRSV epitopes through DNA vaccination can also prime B cells, eliciting a detectable, GP5-specific, humoral immune response [61]. The data obtained showed that both peptides can induce antibody-secreting B cells, which directly correlate with the increase in anti-peptide specific antibodies observed in the IgG evaluation. Surprisingly, the results of primed B cells showed that the group immunized with the peptide GP5-B3 induced a significant increase in primed B cells, greater than that of the group immunized with GP5-B peptide. This could explain why the same level of anti-GP5-B3 IgGs did not change from 21 to 42 dpi. Although these B cells contain the charged antigen in their BCR, they are not activated and therefore have not differentiated into secreting or memory B cells. On the other hand, the group immunized with GP5-B peptide also showed a significant increase compared to the vehicle and control groups, indicating that there are also antigen-charged cells waiting to be activated to differentiate into plasmatic cells or memory cells.

Finally, the evaluation of the subpopulation of naïve B cells showed a decrease in these cells in the groups immunized with peptides, which agrees with the literature. It has been reported that naïve cells are cells that have not been stimulated or come into contact with an antigen; thus, these cells circulate in the bloodstream and lymph nodes waiting to come into contact with an antigen. When naïve B cells encounter an antigen, this subpopulation of B cells diminishes as they differentiate into trained B cells, antibody secretory, or memory cells [62]. Together, these data indicate that the peptides GP5-B and GP5-B3 can stimulate naïve B cells to differentiate into trained cells that will later become antibody-specific secretory B cells or CD2+/CD21− memory B cells.

The PRRSV exhibits significant genetic variability, which influences its interaction with the host’s immune system and the antigenic properties of viral proteins [63]. This virus has a significant capacity to evade the immune response through various mechanisms, including internalization into cells [64]. In contrast, the host response includes neutralizing antibodies against B epitopes in the protein complex involved in the virus´s interaction with the CD163 receptor, which the virus uses to infect [15,64,65].

PRRSV glycoprotein GP5 has been widely studied as a highly antigenic protein and is also the main target of neutralizing antibodies [66]. In fact, the main neutralization epitope is located in a hypervariable region of the ectodomain from the GP5 protein [67]. Thus, vaccines probably have low efficiency in providing protection against homologous wild strains. Other researchers have studied B epitopes from GP5 protein and found that they induce a humoral immune response [16,37,68].

Peptides GP5-B and GP5-B3 were immunoreactive against the sera of vaccinated animals with a commercial vaccine and against sera of animals naturally infected by wild strains, which are largely dominated by genotype 2 strains. However, as the Ministry of Agriculture and Rural Development reported in 2017, circulating strains of PRRSV-1 were also present; so, cross-immunoreactivity against heterologous strains is not ruled out. This can be attributed to the fact that the epitopes contained in the peptides are short sequences that were selected because they were conserved across different in silico reference strains, which was experimentally tested in this immunoreactivity test.

## 5. Conclusions

In conclusion, the data obtained in the present study show that immunization with GP5-B or GP5-B3 peptides induces a sustained humoral response of peptide-specific IgG antibodies. In addition, both peptides induced B cell activation and their differentiation into peptide-specific IgG antibody-forming/plasma B cells. The two peptides, both GP5-B and GP5-B3, were immunoreactive with the sera of animals vaccinated with the commercial vaccine and with the sera from animals naturally infected by circulating strains of PRRSV. These data suggest that B epitopes contained in peptides GP5-B and GP5-B3 are conserved across various strains of PRRSV. This work provides evidence of the capacity of B epitopes in the synthetic peptides GP5-B and GP5-B3 to induce a humoral response. Additionally, it was experimentally demonstrated that these are epitopes preserved in circulating wild strains, which could be included in the future development of a safe and effective vaccine model to prevent PRRSV infection.

## Figures and Tables

**Figure 1 vaccines-12-00652-f001:**
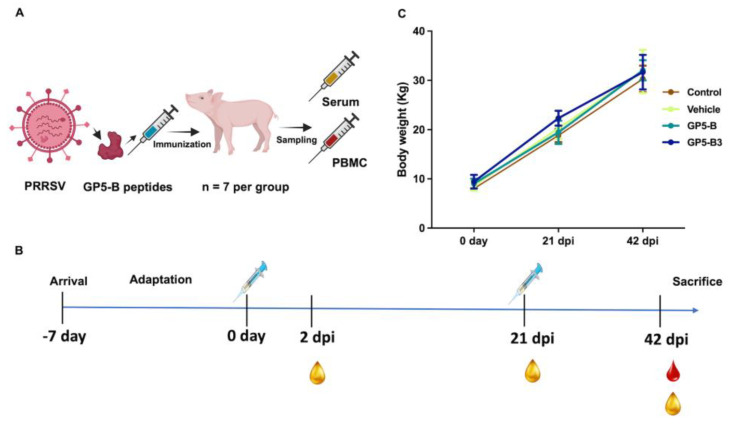
Experimental design and body weight of piglets after immunization. (**A**) Schematic representation of the experimental design; piglets were immunized with GP5-B or GP5-B3 peptides and serum and blood samples were collected. In addition, two control groups (N = 7) were established: a control group and a vehicle group. (**B**) The timeline of the experiment shows the immunization at days 0 and 21, as well as serum sampling at 2 dpi, 21 dpi and 42 dpi. Finally, blood sampling was conducted at 42 dpi. (**C**) Body weight monitoring of piglets in the four groups. (N = 7 per group). The mean weights for the seven pigs in each group are depicted by colored lines, and the interquartile ranges are shown by the error bars. A two-way ANOVA followed by Tukey’s post hoc test was employed to assess statistical significance. The analysis revealed no significant differences in weight between the groups.

**Figure 2 vaccines-12-00652-f002:**
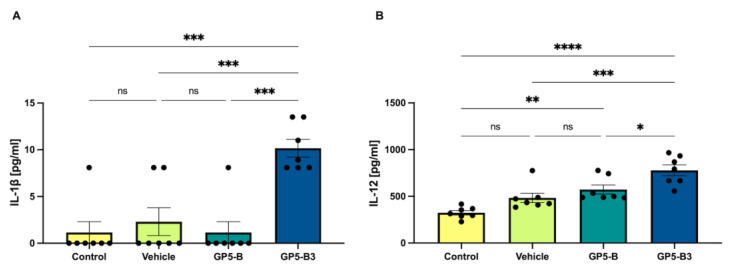
Quantification of IL-1β and IL-12 in pig sera. Sera from piglets were collected at 2 dpi. Cytokines were measured using a multiplex Luminex-based cytokine immunoassay. (**A**) IL-1β. *** *p* < 0.001; ns = not significant. (**B**) IL-12. * *p* < 0.05; ** *p* < 0.005; *** *p* < 0.001; **** *p* < 0.0001; ns = non-significant. Data were presented as means ± SEM. Differences between groups were estimated using one-way ANOVA test with Tukey’s HSD pos hoc.

**Figure 3 vaccines-12-00652-f003:**
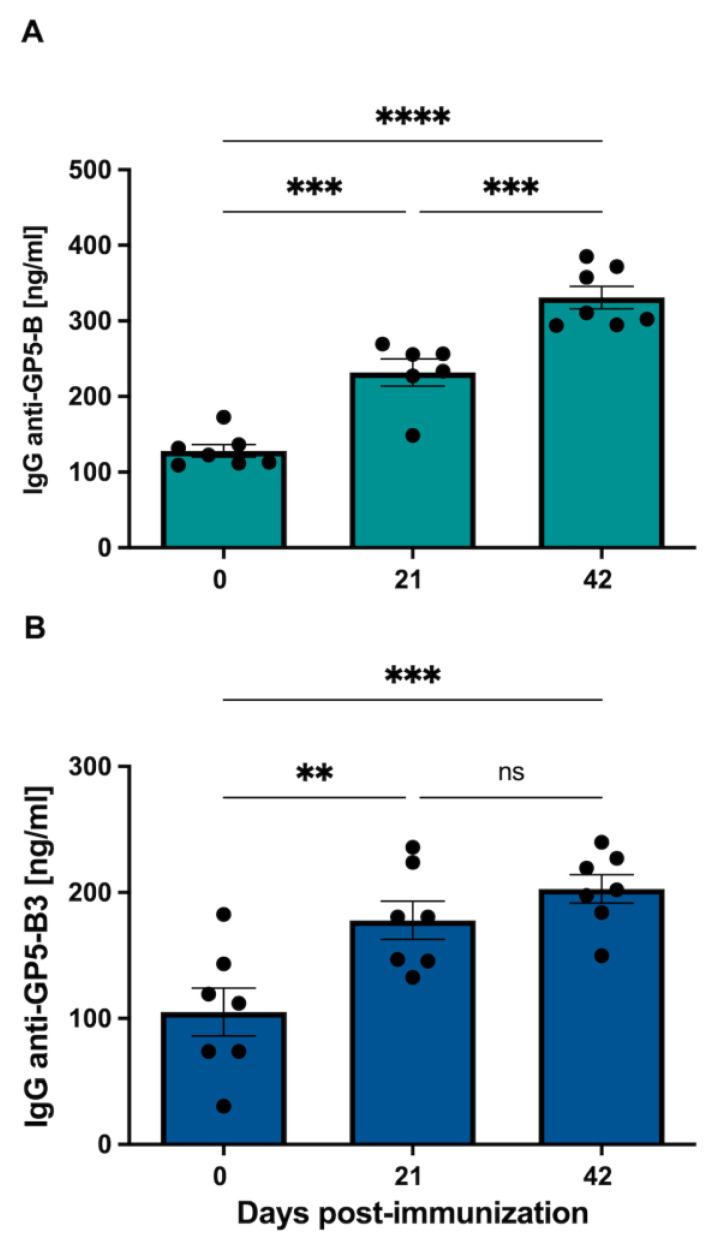
Humoral response mediated by specific anti-peptide IgGs antibodies induced by peptide immunization in piglets. Piglets were immunized intramuscularly (IM) with a peptide coupled to a BSA carrier in each group + adjuvant alhydrogel^®^ (AlOH 2%). The animals were sampled and immunized two times at 21-day intervals. The concentration of IgGs was evaluated at 0, 21 and 42 days dpi through ELISA assay. (**A**) Specific anti-GP5-B IgG. *** *p* < 0.0005, **** *p* < 0.0001. (**B**) Specific anti-GP-B3 IgG. ** *p* < 0.05, *** *p* < 0.001, ns = non-significant. each group. Data were presented as means ± SEM (n = 7). Differences between groups were estimated using one-way ANOVA test with Tukey´s HSD post hoc.

**Figure 4 vaccines-12-00652-f004:**
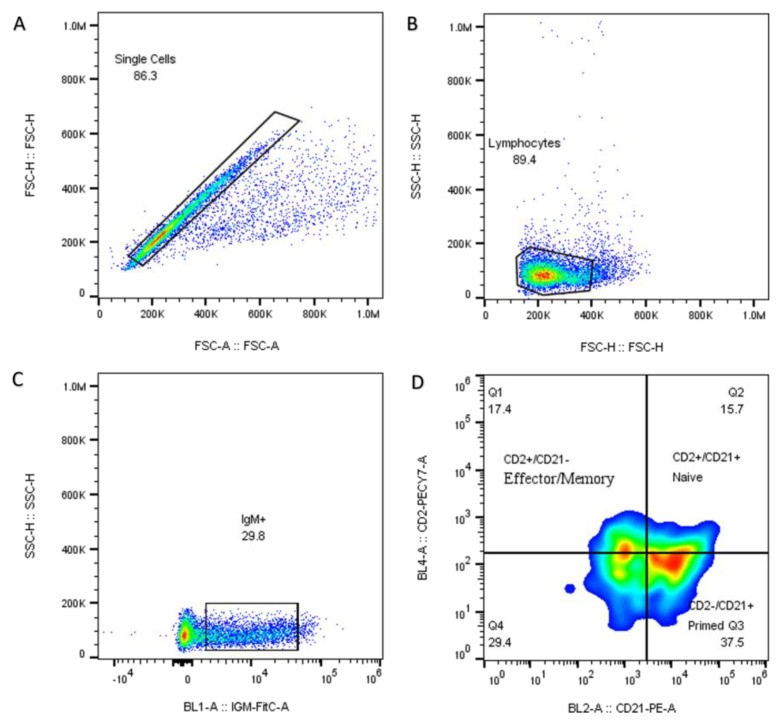
Panel for analysis of B cell subpopulations. Humoral immune response was induced in B cell subpopulations of piglets immunized with synthetic peptides. The cells were recovered at day 42 dpi. The marking was made with anti-CD2 and anti-CD21 markers. (**A**) gating used to select single cells. In (**B**) lymphocyte populations are gated, and B cells were selected by IgM+ marker (**C**). (**D**) the quadrant panel shows the selection strategy for analyzing CD2+/CD21+; CD2−/CD21+ and CD2+/CD21− B cells. The colors represent the population density; blue color shows low density; red color represents the highest cell population density.

**Figure 5 vaccines-12-00652-f005:**
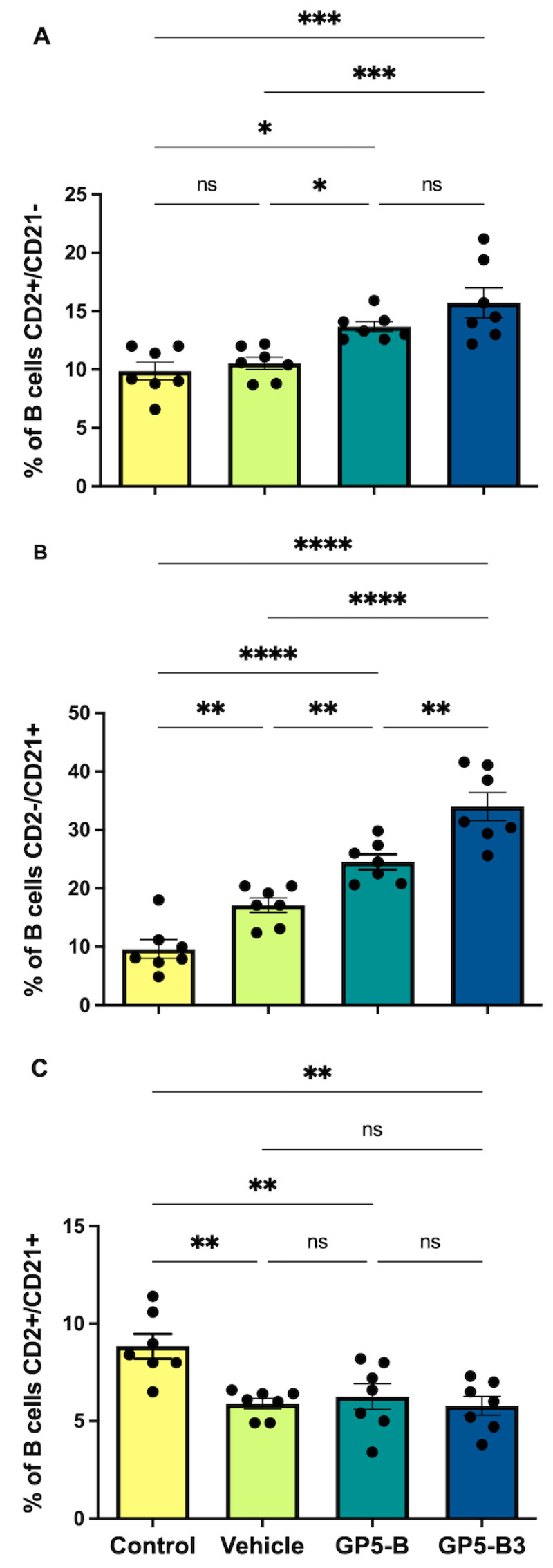
Humoral response mediated by B cell subpopulations in peripheral blood of immunized piglets (MI)**.** Total peripheral blood was recovered through the jugular vein of immunized piglets to isolate mononuclear cells (PBMCs). PBMC isolated from total blood in piglets at 42 dpi, through density gradient with Lymphoprep. A total of 10,000 events were analyzed through flow cytometry by anti-CD2-PECy7, anti-IgM-FITC and anti-CD21-PE. (**A**) Antibody-producing/Memory B cells (CD2+/CD21−) monoclonal antibodies. * *p* < 0.05; *** *p* < 0.0005; ns = not significant. (**B**) Primed B cells (CD2−/CD21+). ** *p* < 0.005; **** *p* < 0.0001. (**C**) Naïve cells (CD2+/CD21+). ** *p* < 0.01. Data were presented as means ± SEM (n = 7). Differences between groups were estimated using one-way ANOVA test with Tukey’s HSD post hoc.

**Figure 6 vaccines-12-00652-f006:**
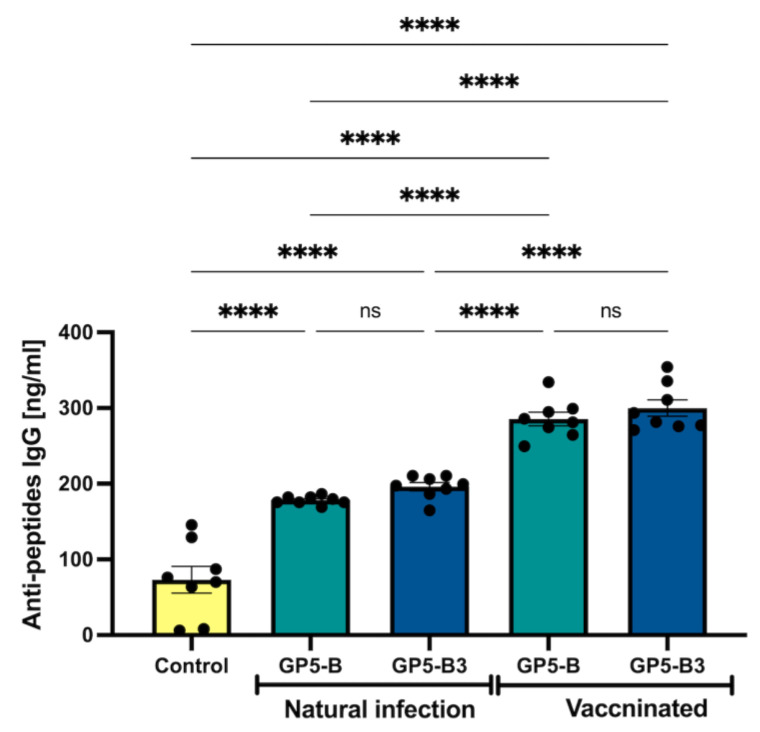
Immunoreactivity between GP5-B and GP5-B3 peptides against the vaccine virus (attenuated INGELVAC-PRRS MLV virus) and circulating field strains. IgG against GP5-B and GP5-B3 were determined in serum of naturally infected animals and in serums of vaccinated IM with a single dose at 0 days. The concentration of IgGs was evaluated at 42 dpi. Anti-peptide antibodies were determined by ELISA assays. The graph shows the recognition of the peptides by IgG antibodies present in the sera of vaccinated animals. Data were presented as means ± SEM (n = 7). Differences between groups were estimated using one-way ANOVA test with Tukey’s HSD post hoc. **** *p* < 0.0001; ns = not significant.

**Table 1 vaccines-12-00652-t001:** Vaccination groups and formulations. The table shows the volumes and concentrations of the components for each formulation applied to each experimental group.

Origin	Farm	Group (N = 7)	Formulation					Commercial Vaccine 104.9DICC50
			PBS (µL)	Carrier Maleimide BSA (µg/µL)	Adjuvant Aluminum Hydroxide (µL)	GP5-B Peptide (µg/µL)	GP5-B3 Peptide (µg/µL)	
El Dapo	La Carreta	Control	800					
		Vehicle	200	200	400			
		GP5-B		200	400	200		
		GP5-B3		200	400		200	
	El Dapo	Vaccinated						2000
El Limon	El Limon	Natural infection						

## Data Availability

Data are contained within this article.

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
