# Peer review of "Synthetic Peptides Elicit Humoral Response against Porcine Reproductive and Respiratory Syndrome Virus in Swine"

_vaccines, 2024, doi:10.3390/vaccines12060652_

Round 1

Reviewer 1 Report

Comments and Suggestions for Authors

The GP5 (glycoprotein 5) of the Porcine Reproductive and Respiratory Syndrome Virus (PRRSV) plays a crucial role in the virus’s replication and its interactions with the host’s immune system. In this study, the authors examined the humoral and B-cell mediated immune responses to GP5 linear epitopes from PRRSV in pigs. The findings revealed increased cytokine concentrations and virus-specific IgG levels following immunization. Notably, the GP5-B and GP5-B3 peptides induced higher IgG levels and increased the presence of plasma and primed B cells, while the number of naïve B cells decreased compared to the control group. Both peptides showed immunoreactivity with antibodies from vaccinated and naturally infected pigs, suggesting their potential in eliciting a protective immune response. However, the study did not conclusively demonstrate that the GP5 peptide alone can induce a protective immune response. An in vitro neutralization test of the antiserum from GP5 peptide-immunized animals would be necessary to confirm this.

Additionally, I have several concerns that need addressing:

1. The criteria for identifying CD2-positive cells are not clearly explained in Figure 4D.

2. It would be clearer for readers if the authors provided examples of cell frequencies from each group.

Author Response

The comments were kept for reference and are depicted in italics. The answers are below each comment/question.

REVIEWER 1

Comments and Suggestions for Authors

Q1.The GP5 (glycoprotein 5) of the Porcine Reproductive and Respiratory Syndrome Virus (PRRSV) plays a crucial role in the virus’s replication and its interactions with the host’s immune system. In this study, the authors examined the humoral and B-cell mediated immune responses to GP5 linear epitopes from PRRSV in pigs. The findings revealed increased cytokine concentrations and virus-specific IgG levels following immunization. Notably, the GP5-B and GP5-B3 peptides induced higher IgG levels and increased the presence of plasma and primed B cells, while the number of naïve B cells decreased compared to the control group. Both peptides showed immunoreactivity with antibodies from vaccinated and naturally infected pigs, suggesting their potential in eliciting a protective immune response. However, the study did not conclusively demonstrate that the GP5 peptide alone can induce a protective immune response. An in vitro neutralization test of the antiserum from GP5 peptide-immunized animals would be necessary to confirm this.

R= We agree with you, it is interesting a in vitro neutralization test of the antiserum from GP5 peptide-immunized animals to emphasize that the GP5 peptide alone can induce a “protective immune response”. In the new version deleted the word “protective” in the abstract.  Nevertheless, the study is interesting because described unprecedented scientific information, particularly the humoral and B-cell mediated immunogenic response induced by GP5 lineal epitopes from porcine reproductive and respiratory syndrome virus. We will consider your recommendation about a in vitro neutralization test in future experiments.   

The results presented in the manuscript demonstrate the immunogenicity of the GP5 peptides. We agree that the data do not conclude that the immunogenic response is protective. In consequence, the word “protective” has been deleted from the abstract.

The results thus far represent an interesting contribution to the understanding of the immunogenic properties of GP5 linear epitopes mediated by IgG antibodies and B cells. Certainly, an in vitro neutralization test will be required to demonstrate a protective response against PRRSV. Furthermore, to validate the potential of these peptides as vaccine candidates, an in vivo challenge must be conducted. These two experiments are not contemplated in the present part of the research project but will be carried out in the near future, as other peptides with different immunogenic characteristics are being tested in early-stage development.

Q2. Additionally, I have several concerns that need addressing:

  1. The criteria for identifying CD2-positive cells are not clearly explained in Figure 4D.

R=We agree with you, in fact, in the new version, particularly in the materials and methods section (line 186 to 197 were revised and improved), we added more information to the criteria for identifying CD2-positive cells. Moreover, more information was added about the quadrant panel designed and bounded, particularly mentioned Fluorescence controls (FMOs).

Q3. It would be clearer for readers if the authors provided examples of cell frequencies from each group.

R= The results section was improved, the percentage of cell frequencies was included throughout the text to make the results clearer for readers.

Reviewer 2 Report

Comments and Suggestions for Authors

Comments on objectives of study. The authors should focus the description of the objectives and extend the respective sentence to make it more precise regarding the work.

Also, please explain clearly in the Introduction, the gap in the literature to be filled by the present study.

Comment on methodology. Please add a new sub-section in M&M to describe separately and in all the details, the controls in the study: animals, samples, materials, procedures etc. Please include details about positive and negative controls. This is an important part of the methodology, and it should be described separately and in detail.

Comment about figures. The visualization in the manuscript is poor. Please enhance the quality of graphs, by explaining better in the legends and by colouring the design.

Comment about tables. The lack of tables is significant and reduces the quality of the presentation.

Comment about overall presentation of the manuscript in general. This is poor in general, rated 2/10, so it must be improved during revision.

Comment about references. Some significant references on the same topic published in 2023 and 2024 have been omitted. The authors must compare their results versus findings of previous workers.

Comment about Discussion. I suggest to divide in two sub-sections.

Overall. The manuscript requires extensive improvement. Re-evaluation is necessary before a final recommendation.

Author Response

The comments were kept for reference and are depicted in italics. The answers are below each comment/question.

REVIEWER 2

Q1. Comments on objectives of study. The authors should focus the description of the objectives and extend the respective sentence to make it more precise regarding the work. Also, please explain clearly in the Introduction, the gap in the literature to be filled by the present study.

R= We agree with you, description of the objective was improved. New references were included as detail below.

New references were included in the introduction

  1. Ruedas-Torres, I.; Rodriguez-Gomez, I.M.; Sanchez-Carvajal, J.M.; Larenas-Munoz, F.; Pallares, F.J.; Carrasco, L.; Gomez-Laguna, J. The jigsaw of PRRSV virulence. Vet Microbiol 2021, 260, 109168, doi:10.1016/j.vetmic.2021.109168.DOI 10.1016/j.prevetmed.2023.105976

  1. Li, R.; Qiao, S.; Zhang, G. Reappraising host cellular factors involved in attachment and entry to develop antiviral strategies against porcine reproductive and respiratory syndrome virus. Front Microbiol 2022, 13, 975610, doi:10.3389/fmicb.2022.975610.

  1. Sun, Q.; Xu, H.; An, T.; Cai, X.; Tian, Z.; Zhang, H. Recent Progress in Studies of Porcine Reproductive and Respiratory Syndrome Virus 1 in China. Viruses 2023, 15, doi:10.3390/v15071528.

  1. Duerlinger, S.; Knecht, C.; Sawyer, S.; Balka, G.; Zaruba, M.; Ruemenapf, T.; Kraft, C.; Rathkjen, P.H.; Ladinig, A. Efficacy of a Modified Live Porcine Reproductive and Respiratory Syndrome Virus 1 (PRRSV-1) Vaccine against Experimental Infection with PRRSV AUT15-33 in Weaned Piglets. Vaccines (Basel) 2022, 10, doi:10.3390/vaccines10060934.

  1. Meng, X.J.; Paul, P.S.; Halbur, P.G.; Lum, M.A. Characterization of a high-virulence US isolate of porcine reproductive and respiratory syndrome virus in a continuous cell line, ATCC CRL11171. J Vet Diagn Invest 1996, 8, 374-381, doi:10.1177/104063879600800317.

  1. Ko, J.H.; Nguyen, P.-L.; Ahn, J.-Y.; Yoon, H.; Min, J.; Lee, L.; Cho, S.-J.; Sekhon, S.S.; Kim, Y.-H. The global research trend of porcine reproductive and respiratory syndrome virus (PRRSV): A mini review. Toxicology and Environmental Health Sciences 2015, 7, 241-250, doi:10.1007/s13530-015-0254-9.

  1. Guo, C.; Liu, X. Editorial: Porcine reproductive and respiratory syndrome virus - animal virology, immunology, and pathogenesis. Front Immunol 2023, 14, 1194386, doi:10.3389/fimmu.2023.1194386.

  1. Garcia Duran, M.; Costa, S.; Sarraseca, J.; de la Roja, N.; Garcia, J.; Garcia, I.; Rodriguez, M.J. Generation of porcine reproductive and respiratory syndrome (PRRS) virus-like-particles (VLPs) with different protein composition. J Virol Methods 2016, 236, 77-86, doi:10.1016/j.jviromet.2016.03.021.

  1. Wissink, E.H.; Kroese, M.V.; van Wijk, H.A.; Rijsewijk, F.A.; Meulenberg, J.J.; Rottier, P.J. Envelope protein requirements for the assembly of infectious virions of porcine reproductive and respiratory syndrome virus. J Virol 2005, 79, 12495-12506, doi:10.1128/JVI.79.19.12495-12506.2005.

  1. Cao, Q.M.; Ni, Y.Y.; Cao, D.; Tian, D.; Yugo, D.M.; Heffron, C.L.; Overend, C.; Subramaniam, S.; Rogers, A.J.; Catanzaro, N.; et al. Recombinant Porcine Reproductive and Respiratory Syndrome Virus Expressing Membrane-Bound Interleukin-15 as an Immunomodulatory Adjuvant Enhances NK and gammadelta T Cell Responses and Confers Heterologous Protection. J Virol 2018, 92, doi:10.1128/JVI.00007-18.

  1. Madapong, A.; Saeng-Chuto, K.; Boonsoongnern, A.; Tantituvanont, A.; Nilubol, D. Cell-mediated immune response and protective efficacy of porcine reproductive and respiratory syndrome virus modified-live vaccines against co-challenge with PRRSV-1 and PRRSV-2. Sci Rep 2020, 10, 1649, doi:10.1038/s41598-020-58626-y.

  1. Kick, A.R.; Amaral, A.F.; Frias-De-Diego, A.; Cortes, L.M.; Fogle, J.E.; Crisci, E.; Almond, G.W.; Kaser, T. The Local and Systemic Humoral Immune Response Against Homologous and Heterologous Strains of the Type 2 Porcine Reproductive and Respiratory Syndrome Virus. Front Immunol 2021, 12, 637613, doi:10.3389/fimmu.2021.637613.

  1. Choi, K.; Park, C.; Jeong, J.; Kang, I.; Park, S.J.; Chae, C. Comparison of commercial type 1 and type 2 PRRSV vaccines against heterologous dual challenge. Vet Rec 2016, 178, 291, doi:10.1136/vr.103529.

  1. Scortti, M.; Prieto, C.; Alvarez, E.; Simarro, I.; Castro, J.M. Failure of an inactivated vaccine against porcine reproductive and respiratory syndrome to protect gilts against a heterologous challenge with PRRSV. Vet Rec 2007, 161, 809-813.

  1. Kim, H.; Kim, H.K.; Jung, J.H.; Choi, Y.J.; Kim, J.; Um, C.G.; Hyun, S.B.; Shin, S.; Lee, B.; Jang, G.; et al. The assessment of efficacy of porcine reproductive respiratory syndrome virus inactivated vaccine based on the viral quantity and inactivation methods. Virol J 2011, 8, 323, doi:10.1186/1743-422X-8-323.

  1. Charerntantanakul, W. Porcine reproductive and respiratory syndrome virus vaccines: Immunogenicity, efficacy and safety aspects. World J Virol 2012, 1, 23-30, doi:10.5501/wjv.v1.i1.23.

  2. Jeong, J.; Park, C.; Oh, T.; Park, K.H.; Yang, S.; Kang, I.; Park, S.J.; Chae, C. Cross-protection of a modified-live porcine reproductive and respiratory syndrome virus (PRRSV)-2 vaccine against a heterologous PRRSV-1 challenge in late-term pregnancy gilts. Vet Microbiol 2018, 223, 119-125, doi:10.1016/j.vetmic.2018.08.008.

  1. Zhao, G.; Zhang, J.; Sun, W.; Xie, C.; Zhang, H.; Gao, Y.; Wen, S.; Ha, Z.; Nan, F.; Zhu, X.; et al. Immunological evaluation of recombination PRRSV GP3 and GP5 DNA vaccines in vivo. Front Cell Infect Microbiol 2022, 12, 1016897, doi:10.3389/fcimb.2022.1016897.

DOI  https://doi.org/10.3390/ani13050813

  1. Liu, D.; Chen, Y. Epitope screening and vaccine molecule design of PRRSV GP3 and GP5 protein based on immunoinformatics. J Cell Mol Med 2024, 28, e18103, doi:10.1111/jcmm.18103.

doi:10.1038/nrd2224

  1. Charerntantanakul, W.; Platt, R.; Johnson, W.; Roof, M.; Vaughn, E.; Roth, J.A. Immune responses and protection by vaccine and various vaccine adjuvant candidates to virulent porcine reproductive and respiratory syndrome virus. Vet Immunol Immunopathol 2006, 109, 99-115, doi:10.1016/j.vetimm.2005.07.026.

New references were included in the discussion section

  1. Pettini, E.; Medaglini, D.; Ciabattini, A. Profiling the B cell immune response elicited by vaccination against the respiratory virus SARS-CoV-2. Front Immunol 2022, 13, 1058748, doi:10.3389/fimmu.2022.1058748.

  1. Buermann, A.; Borns, K.; Römermann, D.; Plege-Fleck, A.; Schwinzer, R. B cell activation and induced antibody responses to porcine antigen can be diminished by PD-L1-mediated triggering of PD-1. Xenotransplantation 2014, 21, 192-192, doi:https://doi.org/10.1111/xen.12083_18.

  1. Mulupuri, P.; Zimmerman, J.J.; Hermann, J.; Johnson, C.R.; Cano, J.P.; Yu, W.; Dee, S.A.; Murtaugh, M.P. Antigen-specific B-cell responses to porcine reproductive and respiratory syndrome virus infection. J Virol 2008, 82, 358-370, doi:10.1128/JVI.01023-07.

  1. Ren, J.; Lu, H.; Wen, S.; Sun, W.; Yan, F.; Chen, X.; Jing, J.; Liu, H.; Liu, C.; Xue, F.; et al. Enhanced immune responses in pigs by DNA vaccine coexpressing GP3 and GP5 of European type porcine reproductive and respiratory syndrome virus. J Virol Methods 2014, 206, 27-37, doi:10.1016/j.jviromet.2014.05.021.

Q2. Comment on methodology. Please add a new sub-section in M&M to describe separately and in all the details, the controls in the study: animals, samples, materials, procedures etc. Please include details about positive and negative controls. This is an important part of the methodology, and it should be described separately and in detail.

R=We agree with you, we added a new table to describe separately and in all the details, the controls in the study: animals, samples, materials, and procedures. We included details about positive and negative controls.

Q3. Comment about figures. The visualization in the manuscript is poor. Please enhance the quality of graphs, by explaining better in the legends and by colouring the design.

R= In the new version graphs and legends were edited to improve the quality.

Q4 Comment about tables. The lack of tables is significant and reduces the quality of the presentation.

R= A table was included to describe the experimental groups in the materials and methods section

Q5. Comment about overall presentation of the manuscript in general. This is poor in general, rated 2/10, so it must be improved during revision.

R=The presentation of the manuscript was improved; the figures were edited to enhance their quality and a table was added.

Q6. Comment about references. Some significant references on the same topic published in 2023 and 2024 have been omitted. The authors must compare their results versus findings of previous workers.

R= New references were added.

  1. Angulo, J.; Yang, M.; Rovira, A.; Davies, P.R.; Torremorell, M. Infection dynamics and incidence of wild-type porcine reproductive and respiratory syndrome virus in growing pig herds in the U.S. Midwest. Prev Vet Med 2023, 217, 105976, doi:10.1016/j.prevetmed.2023.105976.

  1. Sun, Q.; Xu, H.; An, T.; Cai, X.; Tian, Z.; Zhang, H. Recent Progress in Studies of Porcine Reproductive and Respiratory Syndrome Virus 1 in China. Viruses 2023, 15, doi:10.3390/v15071528.

Guo, 2023. DOI 10.3389/fimmu.2023.1194386

  1. Zhao, G.; Zhang, J.; Sun, W.; Xie, C.; Zhang, H.; Gao, Y.; Wen, S.; Ha, Z.; Nan, F.; Zhu, X.; et al. Immunological evaluation of recombination PRRSV GP3 and GP5 DNA vaccines in vivo. Front Cell Infect Microbiol 2022, 12, 1016897, doi:10.3389/fcimb.2022.1016897.

  1. Luo, Q.; Zheng, Y.; Zhang, H.; Yang, Z.; Sha, H.; Kong, W.; Zhao, M.; Wang, N. Research Progress on Glycoprotein 5 of Porcine Reproductive and Respiratory Syndrome Virus. Animals (Basel) 2023, 13, doi:10.3390/ani13050813.
  2. Liu, D.; Chen, Y. Epitope screening and vaccine molecule design of PRRSV GP3 and GP5 protein based on immunoinformatics. J Cell Mol Med 2024, 28, e18103, doi:10.1111/jcmm.18103.

Q7. Comment about Discussion. I suggest to divide in two sub-sections.    

R= We agree that the discussion needed restructuring to improve content, arguments, clarity, and order. Accordingly, we have revised the text and enhanced its content in all the mentioned aspects. Consequently, we believe the new text does not need to be separated into two sections and can remain as a single section.

Q8. Overall. The manuscript requires extensive improvement. Re-evaluation is necessary before a final recommendation.

R= We appreciate your review and recommendations. The observations were addressed and all the manuscript was improved.

Reviewer 3 Report

Comments and Suggestions for Authors

The research question is very important and deserves investigation. The study describes the humoral and B-cell mediated immunogenic response induced by GP5 lineal epitopes from porcine reproductive and respiratory syndrome virus (PRRSV). Pigs were immunized with specific synthetic GP5 peptides and, after developing the immune response, specific IgGs and cross-reactivity against peptides and B-cell subpopulations were analyzed. In conclusion, the study demonstrated that two GP5 peptides elicited immunogenicity mediated by antigen specific IgG´s and B-cell activation.

In my opinion, the study is interesting and may present unprecedented scientific information. The manuscript was relatively well-elaborated and it is possible to understand the scientific background as well as the rationale design of the study. However, it must be reviewed by the authors in some very important details as follows:

1)    Title: it’s too long. An effort to reduce it would be welcome.

2)    Abstract: avoid using the pronoun “we”, especially as the first word. I suggest using passive voice throughout the manuscript. It is also not necessary to explain the methodology in detail. It is more important to highlight the main results in the Abstract. Please present the values (and not only “showed an increase”, “immunized pigs decrease”, “exhibited immunoreactivity”, etc.). Remember: this Abstract will be published in scientific articles databases!   

3)    Introduction: first I recommend a better ordering of the paragraphs, avoiding mixing subjects (as for example in the first paragraph, combining economic importance with viral structure) and paragraphs with just one sentence (lines 61 and 62). Furthermore, it would be very important to describe the structure of PRRSV particles, explaining the main proteins and glycoproteins. Importantly, the study focuses on immunization with a specific viral glycoprotein (GP5), so readers need more information about the viral particle. I also missed a better description of the main commercial vaccines. Explain further the viral genotypes of these commercial vaccines and whether they have already been tested to demonstrate homologous and heterologous protection.

4)    Methodology: it is well described, presenting detailed information. I missed a schematic figure to present the different treatments in section 2.2. Furthermore, three subsections are not necessary to describe blood and serum collection. Please merge them all into one dedicated section. Analytical laboratory procedures could also be reduced to two subsections.

5)    Results: This section is well prepared and organized. But I suggest removing the uninformative Figure 1. The information presented can be described in the text or the Figure can be presented as supplementary material.

6)    Discussion: needs to be better elaborated. It is repeating the presentation of the Results, such as the third paragraph (lines 317 to 323). It is also necessary to remove many other sentences (just two examples: lines 326 to 328 and line 384). The results should be explained in the Results section. Also remove “relevant model” (line 313). Then rewrite the discussion section.

Furthermore, an in-depth review of the English language and scientific nomenclature is necessary (see the correct nomenclature of viruses, including order, family, genus and species names).

Comments on the Quality of English Language

Author Response

The comments were kept for reference and are depicted in italics. The answers are below each comment/question.

REVIEWER 3

The research question is very important and deserves investigation. The study describes the humoral and B-cell mediated immunogenic response induced by GP5 lineal epitopes from porcine reproductive and respiratory syndrome virus (PRRSV). Pigs were immunized with specific synthetic GP5 peptides and, after developing the immune response, specific IgGs and cross-reactivity against peptides and B-cell subpopulations were analyzed. In conclusion, the study demonstrated that two GP5 peptides elicited immunogenicity mediated by antigen specific IgG´s and B-cell activation.

In my opinion, the study is interesting and may present unprecedented scientific information. The manuscript was relatively well-elaborated and it is possible to understand the scientific background as well as the rationale design of the study. However, it must be reviewed by the authors in some very important details as follows:

Q1. Title: it’s too long. An effort to reduce it would be welcome.

R= The title was edited.

Q2.   Abstract: avoid using the pronoun “we”, especially as the first word. I suggest using passive voice throughout the manuscript. It is also not necessary to explain the methodology in detail. It is more  b the main results in the Abstract. Please present the values (and not only “showed an increase”, “immunized pigs decrease”, “exhibited immunoreactivity”, etc.). Remember: this Abstract will be published in scientific articles databases!  

R= The word “we” was edited, the sentence in the abstract was rewritten. New information was added in the abstract, including values such as the mean and standard error in the results description.

Q3.  Introduction: first I recommend a better ordering of the paragraphs, avoiding mixing subjects (as for example in the first paragraph, combining economic importance with viral structure) and paragraphs with just one sentence (lines 61 and 62). Furthermore, it would be very important to describe the structure of PRRSV particles, explaining the main proteins and glycoproteins. Importantly, the study focuses on immunization with a specific viral glycoprotein (GP5), so readers need more information about the viral particle. I also missed a better description of the main commercial vaccines. Explain further the viral genotypes of these commercial vaccines and whether they have already been tested to demonstrate homologous and heterologous protection.

R= The introduction was edited and improved. New information and references were added to address the observation.

 1996. 10.1177/104063879600800317

2005 doi:10.1128/JVI.79.19.12495–12506.2005

2006. 1016/j.vetimm.2005.07.026

2007. https://pubmed.ncbi.nlm.nih.gov/18083979/

2007. doi:10.1038/nrd2224

2010   doi:10.1016/j.virusres.2010.07.030

2011. https://virologyj.biomedcentral.com/articles/10.1186/1743-422X-8-323

2012. doi: 10.5501/wjv.v1.i1.23

2014  https://doi.org/10.1111/xen.12083_18

2014  https://doi.org/10.1016/j.jviromet.2014.05.021

2015. 1007/s13530-015-0254-9

  1. https://doi.org/10.1136/vr.103529

2018. DOI: https://doi.org/10.1128/jvi.00007-18

2018. https://doi.org/10.1016/j.vetmic.2018.08.008Vreman,

2021 http://dx.doi.org/10.1016/j.jviromet.2016.03.021

2020. https://www.nature.com/articles/s41598-020-58626-y

2021. 10.1016/j.vetmic.2021.109168

2021. 10.3389/fimmu.2021.637613

2021. 1016/j.vetimm.2020.110170

2022. 10.3389/fmicb.2022.975610

2022. 3390/vaccines10060934

2022, https://doi.org/10.3389/fimmu.2022.1058748)

2023. org/10.3390/v15071528

2023. DOI 10.3389/fimmu.2023.1194386

2023. 10.1016/j.prevetmed.2023.105976

2023. https://doi.org/10.3389/fcimb.2022.1016897

2023. https://doi.org/10.3390/ani13050813

2023. DOI: 10.1111/jcmm.18103

Q4 Methodology: it is well described, presenting detailed information. I missed a schematic figure to present the different treatments in section 2.2.

R= A new table was added to describe each treatment.

Q5. Furthermore, three subsections are not necessary to describe blood and serum collection. Please merge them all into one dedicated section.

R= The paragraphs was changed to the methodology section. The three subsections that described blood and serum collection in the methodology were rewritten.

Q6. Analytical laboratory procedures could also be reduced to two subsections.

R= The sections corresponding to the laboratory procedures have been rearranged to improve clarity and order.

Q7.    Results: This section is well prepared and organized. But I suggest removing the uninformative Figure 1. The information presented can be described in the text or the Figure can be presented as supplementary material.

R= As recommended, Figure 1 has been improved and now includes a schematic representation of the animal experimental design. The graph depicting the weight monitoring was kept as it is considered important to have a clearer understanding of the experimental groups and their treatments.

Q8.    Discussion: needs to be better elaborated. It is repeating the presentation of the Results, such as the third paragraph (lines 317 to 323). It is also necessary to remove many other sentences (just two examples: lines 326 to 328 and line 384). The results should be explained in the Results section. Also remove “relevant model” (line 313). Then rewrite the discussion section.

R= Your observation is accurate. The discussion has been revised to better contextualize the research findings within the existing literature, allowing for a comprehensive comparison or contrast with previously reported results. We have included key results, albeit not identical to those in the results section, to facilitate the reader’s ability to recall the main findings and effectively relate them to earlier studies.

Q9. Furthermore, an in-depth review of the English language and scientific nomenclature is necessary (see the correct nomenclature of viruses, including order, family, genus and species names).

R= All the manuscript was revised and improved.

Round 2

Reviewer 1 Report

Comments and Suggestions for Authors

The authors have adequately addressed my initial concerns. The decision to postpone the neutralization test until a later study is understandable. Therefore, I recommend accepting the manuscript for publication.

Author Response

We appreciate your review. No comments.

Reviewer 2 Report

Comments and Suggestions for Authors

The authors have improved the manuscript by taking into account all the comments made. No issues could be found in the revised manuscript.

Author Response

We appreciate your review. No comments.

Reviewer 3 Report

Comments and Suggestions for Authors

Comments to the Authors:

As I have already mentioned, the study is interesting and may present unprecedented scientific information. The manuscript was relatively well-elaborated and it presents a good scientific background as well as the description of the rationale design of the study. I suggested the authors reviewing important details overall in the article and they did a good job:

1)    the Abstract is presenting the main findings.

2)    the Introduction is more complete with the explanation of the PRRSV structure.

3)    the Methodology is better presented, including a figure to understand the treatments.

4)    The Results were improved.

5)    The Discussion has been better prepared and is is more interesting.

However, I do not think the main title is good. Previously, I suggested changing it, since it was too long. I recommended an additional effort of the authors aiming to reduce it and the authors changed including an abbreviation (PRRSV). In my opinion, abbreviations must be avoided in a scientific article title. So, I suggest again revising the title by including the complete name of the virus (porcine reproductive and respiratory syndrome virus), removing some unnecessary specifications (induction of specific IgG antibodies and B-Cell mediated immunity) and focusing in the main aim (Swine immunity provided by synthetic peptides against the porcine reproductive and respiratory syndrome virus or immunogenic response elicited by synthetic peptides against the porcine reproductive and respiratory syndrome virus in swine). A close match between title and objective is also very important!

Author Response

Title was improved. New title: “Synthetic Peptides Elicit Humoral Response Against Porcine Reproductive and Respiratory Syndrome Virus in Swine”